# CACE closed: A multiverse examination of the influence of implementation variability on student outcomes in a randomised controlled trial of a universal, school-based social-emotional learning intervention

**Annie O'Brien**[ID]*, **Joao Santos**[ID], **Neil Humphrey, Margarita Panayiotou**

Manchester Institute of Education, The University of Manchester, Manchester, United Kingdom

* annie.obrien@manchester.ac.uk

## Abstract

### Introduction

Amidst calls for more high-quality research to assess the influence of implementation variability on student outcomes in school-based trials using instrumental variable approaches (e.g., complier average causal effect; CACE), unreported researcher degrees of freedom can limit replicability and lead to uncertainty of conclusions. The current study aims to acknowledge and address these limitations using a multiverse framework. We investigate whether, and how, conclusions of a universal social-emotional learning intervention's efficacy are contingent on decisions about how compliance is defined and modelled.

### Methods

Secondary analysis of data from a cluster randomised control trial of the intervention Passport was undertaken, with schools (k = 62, N = 2,425 children) randomly allocated to intervention (k = 33; N = 1,264) or control (k = 29; N = 1,161) conditions. Ten theoretically plausible specifications for the CACE model were identified and pre-registered, including five definitions of compliance (fidelity, dosage, quality, responsiveness, reach) and two compliance thresholds (50th and 75th percentile). Student relational outcomes (bullying, peer support, loneliness) were assessed pre- and post-intervention.

### Results

Multilevel intent-to-treat analysis revealed null intervention effects. Applying a multiverse framework to CACE revealed variation in model results, manifest in entropy values, the precision of confidence intervals and the direction, size and statistical significance of CACE effects. A statistically significant and negative CACE effect was

**Data availability statement:** The data for this study is available on the OSF: 10.17605/OSF.IO/S5PMW.

**Funding:** This work is part of the first author's PhD research, funded by the Kavli Trust (https://kavlifondet.no/en/ [kavlifondet.no]), grant no. Kavli2021-0000000019. The funders had no role in study design, data collection and analysis, decision to publish, or preparation of the manuscript.

**Competing interests:** The authors report no conflict of interest.

found for peer support when compliance was defined by reach using the 75th percentile (β = −.38, 95% CI (−.68, −.08), E = .71, d = −.21), but non-statistically significant intervention effects were observed for the remaining CACE models.

## Conclusions

A multiverse framework enables transparent reporting of analytic uncertainty in evaluations of implementation variability in school-based trials, thereby offering theoretical, methodological and empirical advancements for implementation science. In doing so, it enables us to move from a fragmented view towards a more coherent understanding of complex interventions in real-world settings.

## Pre-registration

www.osf.io/s5pmw.

## Introduction

Amidst rising mental health problems among young people [1], there is robust evidence that school-based social-emotional learning (SEL) programmes can improve children's social-emotional, behavioural, mental health and academic outcomes in the short and long-term [2–4]. However, smaller effect estimates are often reported when interventions are transported from controlled efficacy trials into real-world contexts [5] and may be explained by implementation variability [4,6–8].

Implementation variability is the difference in how a programme is theorised to how it is actioned in practice [6]. It is a multi-dimensional concept [6], such that delivery can vary across the following dimensions: dosage (the number of lessons delivered); fidelity (the key lesson objectives and core components according to the programme manual); quality (how well facilitators deliver programme components and engage recipients); responsiveness (recipient engagement and participation); adaptations (changes made during delivery), and reach (the proportion of recipients present). A review of prevention programmes reported that average effect size estimates were 2–3 times greater when delivered well, whereas under ideal conditions, effect sizes could be improved 12-fold, however full implementation (100%) is rarely reported and/or achieved [6].

The conclusion that "implementation matters" has been corroborated through recent systematic reviews that have revealed a relationship between implementation variability and student outcomes in universal school-based prevention interventions [9,10]. However, further research using high quality methods is required to further understanding of this relationship [9].

Capitalising on recent advancements in implementation science, we employ the instrumental variable approach CACE (Complier Average Causal Effect) to evaluate implementation variability of a universal, school-based SEL intervention 'Passport: Skills for life' (hereafter Passport) [11] on understudied relational outcomes (loneliness, bullying, peer social support). To address limitations of CACE, namely its

reliance on arbitrary decisions (detailed below), we use a multiverse approach to improve transparency and credibility of conclusions [12]. Thus, this research makes theoretical, methodological and empirical advancements to the fields of implementation science and SEL. We now present a three-part discussion of our aims.

## Addressing social relationships through SEL interventions

Loneliness and childhood victimisation are well-established risk factors associated with problematic mental and physical health and academic functioning [13–15]. Loneliness can occur when there is a perceived difference between the actual and desired quantity and/or quality of one's relationships [16] and is associated with long-lasting negative health outcomes for youth [e.g., depression, anxiety; 17,18]. With experiences of loneliness found to peak in early adolescence and schools identified as critical environments for targeting and alleviating loneliness [17], the UK Government's loneliness strategy aims to prevent loneliness in youth by nurturing connected school communities [19].

Indeed, schools have a duty of care to children, to nurture student growth and to prevent bullying behaviours [20]. Bullying is intentional and reoccurring exclusionary or aggressive behaviour over time and in the presence of a power imbalance between perpetrator and victim [21]. It is associated with poorer academic performance and psychological, behavioural, and social maladjustment for both the bully and the victim [13,22,23]. Approximately 35% of 10–15-year-olds in England and Wales were bullied between 2022−23 [24] with a prevalence of 16% reported in Greater Manchester [25].

Peer social support is an important protective factor in child mental health [26], and moderates negative effects of stressful life events (e.g., bullying). It is defined as access to and receipt of emotional or physical support and resources through interactions with those in one's social network [27,28]. Peer relationships become increasingly important during early adolescence, and young people turn primarily to peers for advice or support when experiencing distress [29]. The UK Government thus aim to support the development of quality peer relationships in schools to improve children's mental health [30].

Children who have poor peer relationships (e.g., experience victimisation, social exclusion or lack close friends) are most likely to experience loneliness [17,31–33]. The 'friendship protection hypothesis' suggests that peer relationships protect against negative experiences and their negative outcomes [34] including loneliness, bullying perpetration and victimisation [35–37]. The importance of fostering caring school environments that promote positive peer relationships is therefore critical [17].

Although meta-analytic evidence demonstrates that SEL interventions improve social-emotional skills (effect size [ES] = 0.21–0.57), peer relationships (ES = 0.22) and reduce bullying behaviours (ES = 0.16–0.22) and victimisation (ES = 0.14–0.24) [2,3], the influence of implementation variability on loneliness and peer support remain underexplored [38,39]. Null intent-to-treat (ITT) effects were found when examining the impact of the PATHS SEL programme (Promoting Alternative THinking Strategies) on social support [40], however moderate intervention effects were revealed in moderate-high compliance classrooms (ES = 0.63) [41]. There is also emerging empirical support that school-based interventions targeting social-emotional competencies can alleviate loneliness [42] and the PATHS programme reduced experiences of loneliness among children in the intervention group relative to the control (ES = 1.35–1.68) [39]. The current study aims to contribute to this building evidence base by examining the impact of Passport on these relational outcomes. While a focus on SEL and relational outcomes in this study is therefore advantageous from a developmental perspective, this decision is also pragmatic (this is a secondary data analysis, as detailed in the Methods).

Underpinned by coping theory, Passport primarily aims to help children identify and use positive coping strategies and to offer, seek and accept help [11]. Lessons on 'Relationships' and 'Dealing with Conflict' aim to support children in making friends, dealing with bullying, and use adaptive coping skills to negate feelings of loneliness or rejection that can occur when a friendship ends. Passport is currently implemented in c.115 schools across the UK; evaluating its efficacy in building relation skills in this context is therefore timely [43]. No research to date has examined the effectiveness of SEL

interventions on loneliness and peer victimisation after accounting for implementation variability in a causal framework [9]. The current study aims to address this research gap.

## Implementation matters in school-based trials

The standard approach to evaluating intervention effects is through applying an ITT framework which upholds the bias-protective principle of randomisation ("once randomised, always analysed"), yet accurate causal inference is limited by the assumption of full compliance (i.e., that the intervention group delivered the intervention in its entirety without deviating from the manual/protocol). Unlike pharmaceutical and medical trials where treatment compliance is closely monitored and/or controlled, the assumption does not hold in school-based trials amidst theoretical support for, and empirical evidence of, variation in intervention delivery [2]. Per-protocol and as-treated are often used alongside ITT to address implementation variability, yet these traditional methods produce biased effect estimates (e.g., by excluding data of non-compliers following randomisation and disrupting group equivalence).

A simplistic solution is to focus solely on the ITT effect [44], yet this can lead to biased estimates of true intervention effects [45], which can manifest in two ways. If participants experience adverse intervention effects and consequently discontinue treatment, but participant adherence is not addressed in the analysis, positive intervention effects are overestimated. Conversely, if participants fail to adhere to the protocol despite positive outcomes, intervention effects are underestimated. Thus, considering the average intervention effect in the context of compliance *alongside* the ITT effect (i.e., the average intervention effect across all participants offered the intervention) provides valuable insight into the true intervention impact among those who *received* the intervention.

There is a call for educational research to move "beyond ITT" and use instrumental variable approaches [46] such as propensity score analysis, treatment effect bounding, and CACE [45]. These approaches use a causal framework that specifically models participant compliance and are popular in the medical, pharmaceutical and psychological literature. Though increasingly employed, they are uncommon in school-based trials [47]. This study focuses on the use of CACE [48] which estimates the impact of the intervention among the subpopulation of compliers [45]. It does so by using random assignment as an instrumental variable and by applying assumptions, such as monotonicity and exclusion restrictions, to infer the proportion of compliers. Although CACE is a robust analytic choice [47], its application to education is not without limitations. Identifying and overcoming these limitations is an issue we now address.

## Overcoming limitations to CACE using a multiverse framework

There are a number of limitations to CACE. First, school-based trials typically model compliance using dosage [49,50], as quantity of lessons delivered is considered an appropriate proxy for estimating the impact of both *offering* the intervention (i.e., ITT effect) and intervention *receipt* (i.e., CACE effect) when school attendance is mandatory. However, dose offered and dose received are conceptually distinct [51]. One could therefore argue that *delivering* an intervention session (i.e., dosage) does not guarantee intervention *receipt* if the programme objectives or core components are not delivered (i.e., poor fidelity); if lessons are delivered with poor quality; if students are not engaged or actively participating during lessons (i.e., low responsiveness), or if students are absent (i.e., low reach). The reliance on dosage as the sole indicator of intervention receipt in CACE models is criticised as arbitrary [52], arguably neglecting empirical evidence that interventions can yield positive outcomes even at low dosage levels [53,54]. Our recent systematic review revealed that multiple SEL implementation dimensions (fidelity, dosage, reach, responsiveness, quality) were associated with students outcomes and could therefore be considered as potential indicators of intervention receipt [9].

Second, compliance is modelled as a binary variable in a CACE model, yet defining 'full' compliance of a prevention intervention is significantly challenging. CACE is dependent on the exclusion restriction assumption (non-compliers derive no benefit from the intervention; see Data analysis) which is difficult to validate in the absence of theory-informed compliance thresholds [55] which are largely untested and unknown in psychological and educational interventions [56].

According to the developer of Passport, there is no theoretically informed threshold for optimal programme delivery and empirical research has not established a definition of compliance in the context of this intervention [e.g., 11]. An inductive approach to defining compliance is an alternative solution, but selecting the cut-off point for defining a complier and non-complier is "somewhat arbitrary along the percent compliance continuum" [46, p.22]. Researchers are encouraged to use multiple sensitivity analyses to observe any change across compliance thresholds [46]. Using the 50th and 75th percentile to define compliance is established practice in educational research [41,49,50,57,58].

The selection of a compliance definition and threshold can impact outcomes of the CACE model. Typically, methodological choices available to researchers range in quality, with strong theoretical and/or empirical support guiding selection of a superior approach. However, when there are numerous sound and theoretically justifiable decisions, selecting one over another is arbitrary [12]. Arbitrary data processing decisions can constrain a dataset and subsequently limit the validity of the conclusions drawn from it [59]. To increase transparency of these limitations, we constructed a multiverse of possible modelling choices for our CACE model.

Multiverse analysis systematically explores how different, equally defensible analytic choices impact research findings, to evaluate the robustness of results and transparently convey uncertainty [12]. A multiverse approach will reveal the instability or robustness of results that hinge on these data processing decisions and help to determine the moderating role of each implementation dimension on outcomes, and the level of implementation that yields desired intervention effects, thus addressing multiple and pertinent research priorities [7,8]. Crenshaw and colleagues [60] demonstrate the strength of this framework in a clinical trial for treatment of post-traumatic stress disorder amidst challenges with participant recruitment, differential attrition between conditions, and missing follow-up data. A multiverse approach facilitated examination of the robustness of results that were based upon pre-registered analytic decisions (e.g., ITT) versus alternative analytic choices that addressed unexpected challenges (e.g., as-treated), demonstrating the treatments' efficacy across all analytic universes and thus ensuring confidence in conclusions drawn. Despite its strength in transparently revealing the fragility or robustness of conclusion, multiverse analysis remains underutilised in this field [61]. Our study extends the application of this framework to school-based intervention research.

### The current study

This paper is a response to the call for increased methodological rigor when assessing implementation variability in education research [9]. We employ CACE to model implementation variability within a causal framework [45], a rigorous analytic method, however unreported researcher degrees of freedom can limit replicability and credibility of conclusions [62]. We therefore apply a multiverse framework to transparently report uncertainty [12].

This research thus aims to address the following research questions: (1) What is the impact of Passport on primary school-aged children's relational outcomes (i.e., loneliness, peer social support, and victimisation)? (2) What is the impact of Passport on primary school-aged children's relational outcomes when accounting for intervention compliance? (3) How does the complier average causal effect of Passport on pupil outcomes vary by compliance definition?

## Method

### Design

A two-arm (intervention vs usual practice) parallel cluster randomised controlled trial design was used. The Passport to Success trial took place in mainstream primary schools across Greater Manchester between September 2022–2025. Passport was implemented in Year 5 classrooms of participating schools in the intervention arm during the 2023/2024 academic year. Schools allocated to the control arm continued delivering the usual school provision. Ethical approval for the trial was obtained from the University of Manchester Research Ethics Committee [Ref: 2022-14050-24401]. The trial was prospectively registered with ISRCTN [ISRCTN12875599] and detailed methodology is

available in the published protocol [43]. This paper reports on methods pertaining to the current study only, which was preregistered on the OSF: www.osf.io/s5pmw.

## Sample

Sixty-two schools participated in the current study. Eligible schools were mainstream, non-independent primary schools situated in Greater Manchester, delivering education to Year 4 – Year 6 students and had not previously delivered Passport. The final student sample consisted of 2425 pupils (50.1% male) in 79 classrooms. Students were eligible to take part in the trial if they were in a Year 4 classroom in a participating school in the 2022/2023 academic year, they had not been opted out of the study and they assented to participate. At baseline, the proportion of participating students that were eligible for free school meals (FSM; 34%) and had special educational needs (SEN; 18%) were greater than the national average [63]. Participant demographics are reported in Table 1. All variables, bar FSM, were balanced between trial arms [see 64].

## Procedure

School recruitment commenced 20.09.2022 and concluded 21.03.2023. The trial manager (JS) contacted Local Authorities in Greater Manchester to nominate local schools and received signed Memorandum of Agreement and a Data Sharing Agreement from sixty-two schools.

Parents/carers of Year 4 children received an information sheet and the opportunity to opt their child out of participating in data collection. In line with schools' in loco parentis responsibilities, children opted out of the study in intervention

Table 1. Participant demographic information and descriptive statistics for study variables.

| Variables | Mean (standard deviation) | |
|---|---|---|
| | Passport | Control |
| Student-level (N) | 1264 | 1161 |
| Sex % (male/female) | 50.7/49.2 | 49.5/50.4 |
| FSM % (no/yes) | 70.6/38.2 | 67/32.9 |
| SEN % (no/yes) | 82.4/17.5 | 82.5/17.4 |
| Peer support T0 | 49.75 (9.72) | 48.75 (10.22) |
| Peer support T1 | 49.57 (9.80) | 49.38 (10.19) |
| Victimisation T0 | 49.55 (10.36) | 49.71 (10.50) |
| Victimisation T1 | 50.06 (10.26) | 50.80 (9.41) |
| Loneliness T0* | 4.96 (1.61) | 5.13 (1.69) |
| Loneliness T1* | 4.76 (1.64) | 4.81 (1.64) |
| Teacher-level (N) | 44 | 35 |
| Gender (male/female) | 16/28 | 11/24 |
| Teaching years | 10.8 (9.5) | 12.6 (11.3) |
| Stress T0* | 7.08 (2.2) | 7.56 (1.56) |
| Coping with stress T0 | 6.39 (2.45) | 6.03 (2.16) |
| Attitudes towards SEL T0 | 49.6 (4.7) | 50.28 (4.7) |
| Dosage (%) | 92.67 (19.28) | – |
| Fidelity (%) | 87.99 (16.25) | – |
| Reach (%) | 89.44 (16.05) | – |
| Responsiveness (%) | 82.08 (19.43) | – |
| Quality of delivery (%) | 88.38 (14.41) | – |

Note. T0 = pre-intervention; T1 = post-intervention. *Higher scores indicate negative outcomes.

schools still received the Passport programme, which was delivered as part of the school curriculum. Participating children assented prior to completing baseline outcome surveys (T0; April – July 2023), administered via Qualtrics.

Randomisation took place following baseline survey data collection by an independent statistician with schools as the unit of randomisation. The allocation sequence was generating using MinimPy software [65]. Minimisation was used to ensure balance across trial arms for school size and proportion of students eligible for FSM.

Year 5 teachers in intervention schools received mandatory 2-hour online training in delivering Passport from Partnership for Children in September 2023; a 1-hour optional booster training session was offered in February 2024 and aimed to address any challenges that arose during the initial months of programme delivery. Post-intervention data collection took place in April – July 2024 (T1).

### Intervention

The reporting of the intervention is guided by the TIDieR checklist [66]. Passport follows the adventures of two children as they navigate a fantasy world with dragons and mythical creatures. There are 18 sequential lessons (approx. 55 minutes) delivered once weekly on topics of emotions, relationships and helping others, dealing with difficult situations (e.g., bullying), fairness and justice, and coping with change and loss. Passport's theory of change posits that fostering coping flexibility will help children self-regulate, inhibiting the onset of mental distress [11]. The programme's logic model is illustrated in Humphrey et al. [64].

Lessons are student-led and delivered during class time by Year 5 teachers using physical and digital resources [67], including comic strips, board games and role play. Teachers must adhere to the structure, sequence and content of the programme manual however they are encouraged to *augment* lessons, personalising them as necessary to meet the individual needs of their class.

### Usual practice

Schools allocated to the control arm of the trial continued with usual practice. A usual practice survey administered to all participating classroom teachers revealed that approximately two thirds of the topics covered by Passport were taught across participating schools (e.g., identifying and managing emotions) and two thirds of classrooms delivered propriety SEL curricula (e.g., Jigsaw).

### Outcome measures

**Student outcomes.** The self-report UCLA loneliness scale for children, recommended by the Office for National Statistics [68], captures loneliness via peer exclusion and isolation (e.g., 'How often do you feel that you have no one to talk to?') using three response categories (e.g., hardly ever or never, some of the time, often). The scale is validated among 10–15-year olds [68] and has strong internal reliability ($\alpha = 0.87 - 0.89$) [69,70]. Alpha for the current sample at T0 was .71. Scores are summed, with higher scores indicating greater experiences of loneliness.

Peer victimisation was assessed using the 3-item social acceptance dimension of the KIDSCREEN-52 (KS52) health related quality of life (HRQoL) measure [e.g., 'Thinking about the last week, have other girls and boys made fun of you?'; 71]. Items are reverse scored, with higher scores indicating lower bullying incidence. The scale has strong internal reliability ($\alpha = 0.77$) and convergent validity [Pearson's $r = 0.29$–$0.32$; 71]. Internal consistency for the current sample at T0 was .75.

Peer support was measured with the 4-item Social Support and Peers dimension of the KIDSCREEN-27 (KS27) HRQoL [72]. The scale captures participant's relationships with other children (e.g., 'Thinking about the last week, have you been able to rely on your friends?'), has strong test-retest reliability ($ICC = 0.61$), criterion validity ($r = 0.94$) and convergent validity [$r = 0.36$; 72]. Alpha for the current sample at T0 was .71. Scores are summed, and higher scores indicate positive peer relationships.

Both the KS27 and KS52 are self-report measures with 5 response items (e.g., Never, seldom, quite often, very often, always) and have been cross-culturally validated among young people aged 8–18-years [71,73]. Scores were adjusted for gender and age, following the manual guidelines.

**Compliance definition.** Implementation data was collected at a single time point (T1) using a teacher self-report survey (see S1 File). Previous conceptual and empirical work informed survey development [e.g., 6,38] which was reviewed by the research team prior to distribution. Survey items captured dosage (n = 18), fidelity (n = 4), quality of delivery (n = 4), student responsiveness (n = 3), and reach (n = 1). Total scores for each dimension were multiplied by the number of sessions delivered by the respective staff to yield dosage-adjusted implementation scores for each dimension, thus accounting for programme exposure.

**Covariates and compliance predictors.** The approach to selecting the covariates and compliance predictors is detailed in S2 File. Baseline scores across loneliness, peer support and victimisation were included as child-level covariates, and the following data provided by schools included as student-level compliance predictors in the CACE models: pupil FSM eligibility (yes/no), special educational needs status (SEN; yes/no) and sex (male/female).

A staff survey, issued at baseline, included single items to capture perceived stress ('How stressful is your job?') and coping with stress ('How well are you coping with the stress of your job right now?') using an 11-point Likert scale, ranging from not stressful/not well, to very stressful/very well, respectively. Both items have good concurrent and predictive validity [74]. Teacher attitudes towards SEL were measured using the Teacher SEL Beliefs Scale, which includes 12 items (e.g., 'I want to improve my ability to teach social and emotional skills to students') and a 5-point Likert scale (1 = strongly disagree; 5 = strongly agree).

## Data analysis

**CACE overview and assumptions.** Angrist and colleagues [75] described 4 patterns of participant compliance behaviour in the context of binary condition assignment (T; 0 = control, 1 = treatment) and treatment received (D; 0 = not received, 1 = received) for an individual $i$. There are compliers (those who adhere to their condition assignment; $D_i(T_i = 1) = 1$, $D_i(T_i = 0) = 0$); defiers (those who do not adhere to their condition assignment; $D_i(T_i = 1) = 0$, $D_i(T_i = 0) = 1$); always-takers (i.e., those who seek out and receiving treatment despite condition assignment; $D_i(T_i = 1) = D_i(T_i = 0) = 1$) and never-takers (those who do not take the treatment regardless of condition assignment; $D_i(T_i = 1) = D_i(T_i = 0) = 0$). Satisfying the following four assumptions allow us to focus on two groups in the CACE model: compliers and non-compliers only.

These assumptions are (1) independence of intervention assignment and outcomes; (2) individual outcomes are not impacted by the condition assignment of other participants (i.e., the Stable Unit Treatment Value Assumption [SUTVA]) [75]; (3) monotonicity, which assumes there are no defiers and there are at least some compliers, and (4) the exclusion restriction, which assumes that the intervention had zero effects for the non-compliers across the intervention and control arm.

The first two assumptions are satisfied through cluster randomisation, reducing the likelihood of participant interaction (or contamination) across condition assignments. Monotonicity is guaranteed in the current study as schools randomised to the control arm of the trial do not have access to the programme materials [45], and providing the opportunity to participate ensures that some participants will actually do so [44]. Satisfying monotonicity allows clear classification of those who do not adhere to condition assignment in the intervention arm as never-takers, and the control arm as always-takers thus removing 'defiers'.

The exclusion restriction is difficult to guarantee in school-based trials, as non-compliance is rarely zero, and partial implementation may still yield benefits. This contrasts with medical trials, such as vaccine studies, where the exclusion restriction is more plausible since there is a clear binary distinction; participants in the intervention arm who refuse a vaccine (non-compliers) cannot experience immunological benefit and the assumption of zero effect for non-compliers is more credible. Methods to overcome potential violation of this assumption include relaxing of assumption parameters

using strong compliance predictors [76] and using sensitivity analyses when dichotomising a continuous measure of compliance to facilitate comparison of compliance thresholds [45]. With no defiers and no intervention effect for never- or always-takers, participants can be dichotomised into compliers vs non-compliers.

Structural equation mixture modelling (SEMM) provides a probabilistic framework to estimate the CACE effect. SEMM assumes that the sample is made up of unobservable latent subgroups (compliers, non-compliers). Compliance behaviour (captured via the self-report implementation survey) is observed in the intervention arm of an RCT and is used, alongside information from the outcome variables and compliance predictors, to estimate the probability of each participant in the control group belonging to a latent class; this framework identifies those in the control arm who would be compliers (had they been offered the intervention) [46]. In other words, the two latent classes include participants from both the intervention (based on observed compliance) and control (based on probabilistically estimated compliance) groups to allow for the comparison of intervention compliers vs. control "would be" compliers. Assuming exclusion restriction, the comparison between intervention non-compliers and control would be non-compliers is set to zero (no intervention effects assumed for those that did not comply).

**Selecting a compliance definition and threshold.** Construction of the multiverse data set is detailed in S2 File and visually illustrated in Fig 1. Ten theoretically plausible specifications were identified and pre-registered (https://osf. io/z62wa). Skewed dosage data resulted in 9 specifications (see Compliance Thresholds) and thus 27 models were analysed. The first author was responsible for data analysis and accessed the data following publication of the pre-registration.

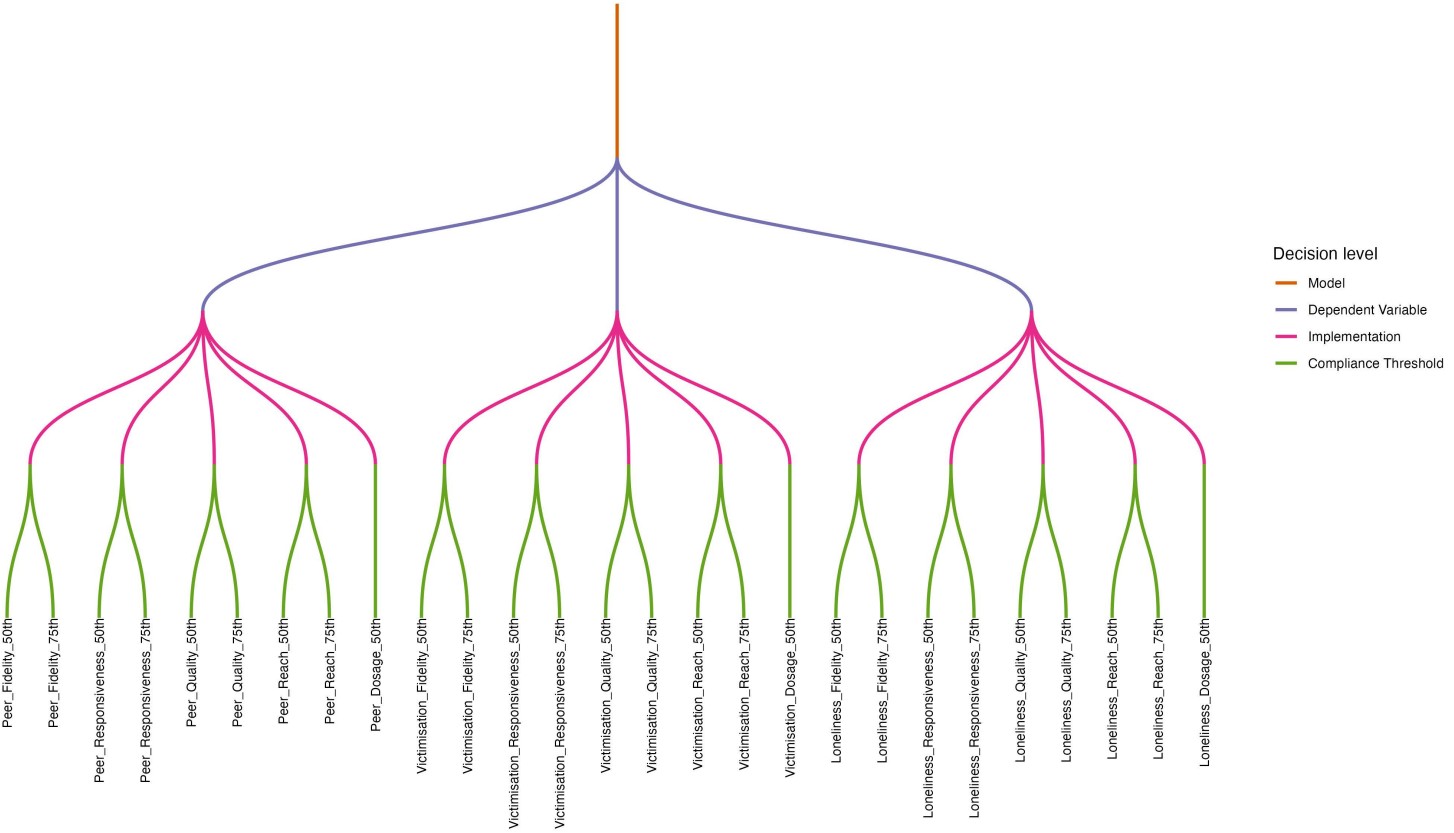

**Fig 1. Decision tree for multiverse specifications.**

**Compliance definition.** Five theoretically plausible approaches to defining compliance were identified: fidelity, dosage, quality, responsiveness and reach. Each were introduced in separate CACE models.

**Compliance thresholds.** In line with recommendations [46] and established practice [41,49,57,58], two compliance thresholds were used to define compliers and non-compliers: the 50th and 75th percentiles. This decision was also pragmatic, as each CACE model requires substantial computational time (>24 hours) and it will facilitate comparison with other intervention studies using identical thresholds, such as the ones cited above. Classrooms that fell below the 50th percentile were classified as non-compliers, whereas classrooms above the 50th percentile were defined as moderate compliers and those above the 75th percentile as high compliers. Due to high levels of dosage, the 50th percentile only was used for compliance defined by dosage.

**Statistical analysis.** Statistical analyses were performed using Mplus Version 8 [77] and results were summarised via the Mplus Automation and ggplort in R Version 4.0.5 [78]. For code, see https://osf.io/z62wa. While the intervention was randomised at the school level, three-level mixture modeling is currently not possible. We therefore regressed y on x using two-level mixture hierarchical linear models (level 1 = child; level 2 = classroom) with random intercept fixed slopes and standard errors (SE) adjusted to account for school-level clustering (TYPE = COMPLEX). Classroom, and not school, was chosen as the level 2, as programme delivery occurred at the classroom-level. The compliance latent variable was estimated with MLR estimation and expectation-maximisation (ML-EM) algorithm [79].

Student-level covariates (baseline scores, sex, SEN) were modelled at the within-level and teacher-level covariates (stress, coping, attitudes) at the between-level. Full information maximum likelihood (FIML) was used to handle missing data. A correction for multiplicity was not applied in the current study as the multiverse is exploratory, in contrast with specification curve analysis for hypothesis testing [80]. The goal is not to correcting for multiplicity (i.e., alpha adjustment) but to increase transparency by visualising the distribution of outcomes [12].

## Results

Descriptive statistics for study variables are reported in Table 1. Psychometric properties for all variables are reported in S3 File. Model fit was acceptable across all variables however the teacher sample size was small, and psychometric testing should be replicated with a larger sample.

### ITT analyses

After controlling for student and classroom-level covariates, null ITT effects were found for peer support, victimisation and loneliness (see Table 2).

### CACE analyses

The CACE effects for each analytic universe are reported in Table 3 (for individual model results, including entropy values, class count, and strength of predictors, see S4 File). The entropy value (E; range 0–1) indicates how well a model distinguishes between latent classes, with high values indicating stronger class separation between compliers and non-compliers [81]. Entropy values varied across models and only four met the pre-registered threshold of .76 [82]. The four

Table 2. Standardised ITT effect for relational outcomes (N = 2,425).

| Predictor | Outcome variables β (SE) | | |
|---|---|---|---|
| | Peer support | Victimisation | Loneliness |
| **Passport vs Control** | −.02 (.16) | −.21 (.15) | .02 (.16) |
| | 95% CI (−0.29, 0.25) | 95% CI (−0.44, 0.03) | 95% CI (−0.23, 0.28) |

**Table 3. Standardised CACE effect for relational outcomes (N = 2425).**

| Predictor | Compliance definition | Outcome variables β (SE) *Cohen's d* | | | | | |
|---|---|---|---|---|---|---|---|
| | | Peer support | | Victimisation | | Loneliness | |
| **Passport vs Usual Provision** | | | | | | | |
| | Moderate fidelity | .32 (.3) | *.13* | .18 (.68) | *.08* | −.07 (.36) | *−.02* |
| | High fidelity | −.49 (1.25)✢ | *−0.26* | .24 (.21) | *.12* | −.39 (.27) | *−.14* |
| | Moderate dosage | .22 (.24) | *.09* | .23 (5.6) | *.08* | −.15 (.21) | *−.05* |
| | Moderate responsiveness | −.33 (2.7)✢ | *−.14* | .38 (.24) | *.14* | −.04 (.38) | *−.02* |
| | High responsiveness | −.38 (.24) | *−.19* | −.31 (.12) | *−.18* | −.08 (.23) | *−.04* |
| | Moderate quality | .31 (.29) | *.13* | .35 (.24) | *.12* | −.05 (.23) | *−.02* |
| | High quality | −.38 (.24) | *−.23* | .4 (.29) | *.13* | −.12 (3.7) | *−.05* |
| | Moderate reach | .4 (.31) | *.17* | .36 (.27) | *.13* | −.08 (.32) | *−.02* |
| | High reach | **−.38 (.18)*** | ***−.21*** | −.41 (.38) | *−.3* | .69 (.44) | *.32* |

Note. These are the standardised model results. 50th percentile = moderate compliance; 75th percentile = high compliance. Positive values indicate improved scores for peer support and victimisation while negative values indicate poorer scores; the opposite is true of loneliness.

\* p < 0.05.

✢ Models without statistically significant compliance predictors.

optimal-entropy models examined intervention effects on victimisation with compliance modelled using moderate reach, engagement, adherence, and dosage. Compliance predictors were weak across multiple CACE models (see S4 File).

The multiverse analyses revealed a statistically significant and *negative* intervention effect for peer support when compliance was defined by high reach (β = −.38, 95% CI (−.68, −.08), E = .71, d = −.21). The remaining 26 CACE models were not statistically significant, indicating that the intervention did not improve student self-report victimisation, loneliness or peer support relative to the control, irrespective of implementation variability. Additionally, several models revealed substantial effect sizes with corresponding unstandardised models considered statistically significant, explained by differences in the sampling distribution between standardised and non-standardised coefficients. Our primary interpretation and reporting are based on standardised results as per our pre-registration, and standardised estimates are reported in S4 File. The results of the multiverse are illustrated by Figs 2–4 for victimisation, loneliness and peer support, respectively.

**Victimisation.** The strongest entropy values were observed across models for victimisation (.7 −.78). CACE effects ranged between −0.41 and 0.4, with seven of nine estimates above zero and all with confidence intervals including zero. Results for moderate compliers were relatively consistent across implementation dimensions, with estimates clustering between 0.18 and 0.38. For high compliers, results showed greater variability depending on how compliance was defined; students in high compliance classrooms were more likely to report poorer scores when compliance was defined by reach (β = −.41, 95% CI (−1.03,.22), E = .73) or engagement (β = −.31, 95% CI (−.63,.01), E = .7), while high quality resulted in the largest positive effect (β = .4, 95% CI (−.06,.86) E = .76). Confidence intervals are wide for dosage compared to other implementation dimensions (β = .23, 95% CI (−8.91, 9.37) E = .78), indicating greater uncertainty in this estimate.

**Loneliness.** Entropy values were lowest for loneliness models (.59 −.69). Results for moderate and compliers were relatively consistent across all but one model, with estimates ranging between −.39 and −.08, indicating small but non-statistically significant improvements in scores. Students in classrooms with high levels of reach however were more likely to report higher levels of loneliness (β = .69, 95% CI (−.04, 1.42), E = .69, d = .32). Confidence intervals are wide when compliance is defined by quality using the 75th percentile (β = −.11, 95% CI (−6.21, 6), E = .65) compared to other implementation dimensions.

**Peer support.** The greatest variation in results is observed for peer support, with one model considered statistically significant when compliance was defined using high reach (β = −.38, 95% CI (−.68, −.08), E = .71, d = −.21). Across

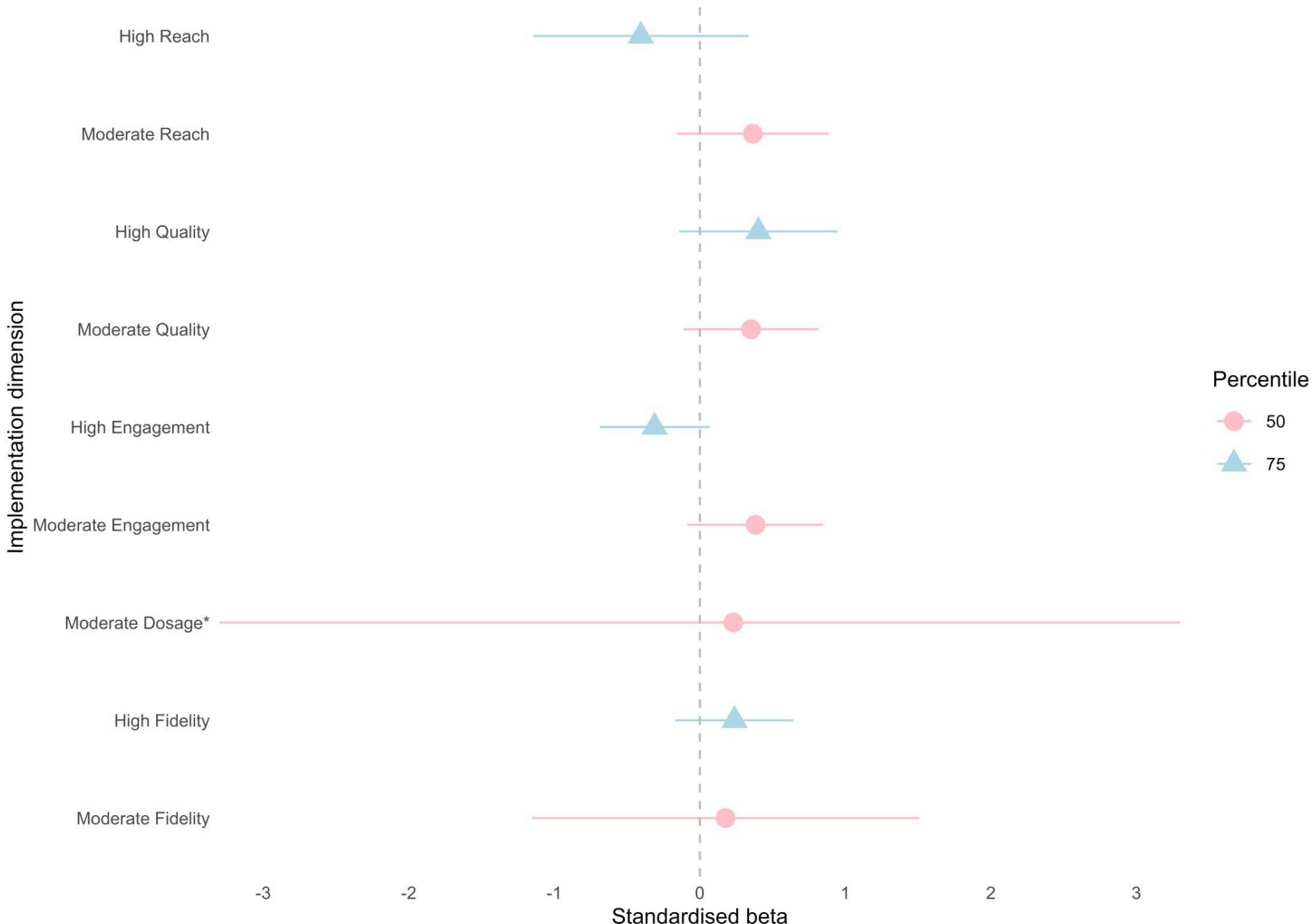

**Fig 2. CACE estimates (x-axis) across five compliance definitions (y-axis) for victimisation outcome.** Compliance definition is along the y-axis and point estimates are plotted as standardised beta coefficients on the x-axis, with pink circles representing the 50th percentile (moderate compliance) and blue triangles representing the 75th percentile (high compliance) across model specifications. Horizontal lines indicate 95% confidence intervals. An asterisk (*) indicates a confidence interval that extends beyond the x-axis limits shown and is truncated for visualisation.

models, estimates varied considerably based on compliance levels and dimensions. For high compliers, estimates cluster between −.49 and −.38 while moderate compliers showed greater range of estimates (−.33 and .4); when the estimate for moderate engagement (the only negative effect; β = −.33) was removed, all estimates were above zero. Within compliance levels (moderate, high), estimates were relatively consistent except the outlier of moderate engagement, however meaningful differences are observed between compliance levels. Negative estimates were observed for high compliers while moderate compliers showed positive estimates, indicating the direction of intervention effects vary depending on the compliance threshold applied. When compliance was modelled by engagement, the difference between high (β = −.38, 95% CI (−.77, .01), E = .74) and moderate compliers (β = −.33, 95% CI (−4.88, 4.23), E = .6) became negligible, although the width of the confidence intervals increased. Similarly, wide intervals are observed for high adherence (β = −.49, 95%

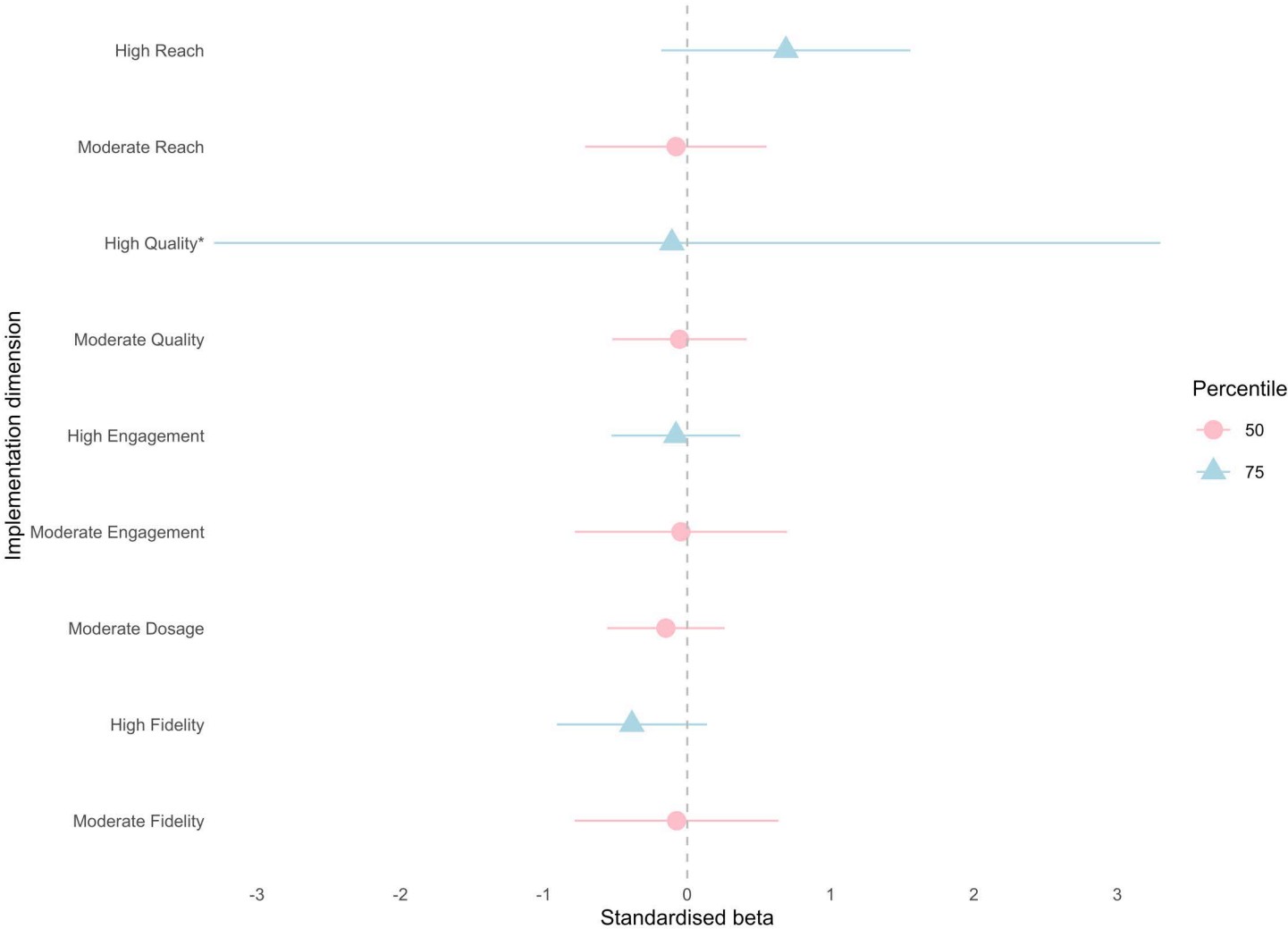

**Fig 3. CACE estimates (x-axis) across five compliance definitions (y-axis) for loneliness outcome.** Compliance definition is along the y-axis and point estimates are plotted as standardised beta coefficients on the x-axis, with pink circles representing the 50th percentile (moderate compliance) and blue triangles representing the 75th percentile (high compliance) across model specifications. Horizontal lines indicate 95% confidence intervals. An asterisk (*) indicates a confidence interval that extends beyond the x-axis limits shown and is truncated for visualisation.

CI (−2.54, 1.56), E = .67). Despite the lack of statistical significance, models showed non-trivial effect sizes for high adherence (d = −.26) and quality (β = −.38, 95% CI (−.77, .01), E = .74, d = −.23).

## Discussion

Amidst calls to use instrumental variable approaches, such as CACE, to address implementation variability in school-based trials, arbitrary analytic decisions can influence results and subsequent conclusions [9]. The current study provides a first step in addressing these limitations using a multiverse framework. We investigated whether, and how, conclusions of an intervention's effectiveness were contingent on decisions about how compliance was defined and modelled. We recommend caution when interpreting the results of the multiverse, which does not identify superior analytic decisions

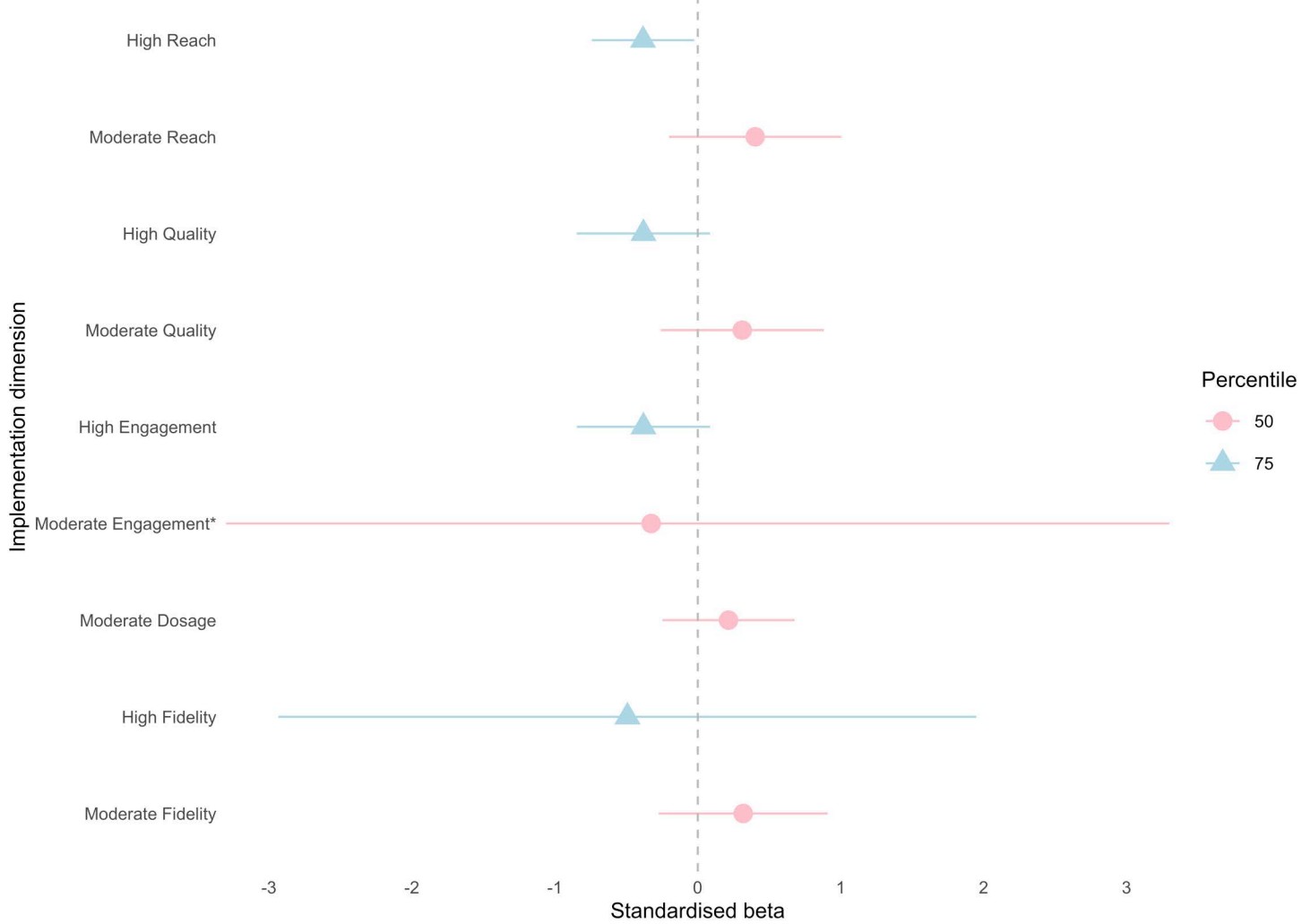

**Fig 4. CACE estimates (x-axis) across five compliance definitions (y-axis) for peer social support outcome.** Compliance definition is along the y-axis and point estimates are plotted as standardised beta coefficients on the x-axis, with pink circles representing the 50th percentile (moderate compliance) and blue triangles representing the 75th percentile (high compliance) across model specifications. Horizontal lines indicate 95% confidence intervals. An asterisk (*) indicates a confidence interval that extends beyond the x-axis limits shown and is truncated for visualisation.

based on, for example, statistical significance, rather it provides a descriptive report of the robustness and replicability of results [12]. In other words, its purpose is not to discover which model conditions result in desirable outcomes (i.e., cherry picking), but transparently report whether results are robust to these variations.

The multiverse analysis revealed consistent results when examining the impact of intervention compliance on loneliness. However, results for victimisation were somewhat inconsistent across analytic universes depending on the compliance definition used. Results for peer support showed the greatest variability across universes, with effects dependent on both the moderator used and compliance threshold applied. Across the multiverse, variation manifested in entropy values, the direction and strength of effects, and CI precision. While CACE effects were sensitive to our choice of compliance definition and threshold, we can confidently conclude that the Passport intervention did not improve students' relational

outcomes relative to the control group, even when accounting for implementation variability. To explore the implications of this variability for intervention research, we examine each in turn.

### Evaluating entropy values

Entropy values ranged between .59–.78 across models with most (85%) below the recommended threshold of .76 [82]. We initially hypothesised that suboptimal entropy values could be explained by negatively skewed dosage data resulting in low score variance across compliance definitions which were weighted by dosage. Transforming continuous implementation dimensions into a binary variable weighted by dosage may result in equifinality [83], whereby different combinations (e.g., low attendance and high dosage vs high attendance and low dosage) produce the same overall compliance score. This is in contrast to a 'clean' binary variable for dosage. However, the multiverse revealed variation across all compliance definitions and thresholds bar dosage, making this theory less viable. A clear pattern emerged when looking at the outcome variables, with the strongest entropy values observed for victimisation models and the weakest for loneliness. Although this hypothesis is speculative, this trend may be explained by the compliance predictors in the CACE models, which included the baseline score of the outcome under investigation. It is possible that bullying victimisation (and by proxy, perpetration) is a stronger predictor of teacher compliance as an externalising behaviour relative to feelings of loneliness which are internalising [84].

### Evaluating precision and confidence intervals (CI)

The precision of point estimates varied across implementation dimensions and outcomes, as reflected in the differing widths of CI. Specifically, wide CI were observed when modelling dosage (victimisation), high quality (loneliness) and moderate engagement (peer support), indicating greater uncertainty around the estimated effects in these models. To a lesser extent, this was also observed for high adherence in relation to peer support.

These patterns may result from a number of potential factors. First, weak predictor variables may contribute to this imprecision. Indeed, some model results should be interpreted with a degree of caution, as the multiverse revealed differences in the strength and specificity of the compliance predictors across CACE models, which also likely impacted the specificity of class separation [85]. Strong compliance predictors are essential to enhancing the reliability and robustness of CACE models, as they directly influence class assignment precision and, consequently, the validity of estimated intervention effects [44]. Model results with weak compliance predictors potentially violate the exclusion restriction assumption and should therefore be interpreted with caution [86].

Despite theory and evidence informing covariate selection, this secondary data analysis was limited to available predictors. This issue of low predictive power of carefully chosen compliance predictors is not unique to the current study [87,88]. Predictors were consistent between models to limit inflating the number of models in the multiverse, nonetheless reviews suggest that contextual and individual factors may differentially influence implementation behaviours in school-based social-emotional and mental health interventions [89,90].

Furthermore, three models with low entropy values had comparatively larger CI: loneliness moderated by high quality (E = .65) and social support moderated by moderate engagement (E = .6) and high adherence (E = .67). However, wide CI for a model with strong entropy (victimisation moderated by dosage; E = .78) requires consideration of an alternative explanation. Ninety-two percent of all lessons were delivered on average resulting in negatively skewed dosage data and a small number of non-compliers (38%, n = 921), reducing power to detect moderation effects. Wide CI therefore reflect limited variability in the moderator, rather than outcome variability. Finally, implementation data may have been subject to measurement error, which is explored further below. Taken together, caution is warranted in interpreting these specific findings as the low precision limits confidence in the estimated effects, discussion of which we now turn.

## Evaluating the direction and size of effects

The CACE effects were largely consistent with the ITT effect, indicating that there was no statistically significant difference between Passport and usual provision for student's relational outcomes for all but one model. Additionally, the multiverse revealed differences in the direction of point estimates between compliance thresholds. When compliance was modelled using the 50th percentile as the cut-off point between compliers and non-compliers, the direction of effects was mainly positive, but direction of effects was mainly negative for high compliance. An interesting trend was also observed for the size of effects: negligible effects were observed for loneliness across all models, except for high reach which had the largest effect size of all 27 models (d = .32). The second largest effect for high reach and victimisation, but effects were negative (d = −.3). Similar effects were observed for peer social support in classrooms with high adherence (d = −.26), high quality (d = −.23) and high reach (d = −.21, p < .05).

Opposing directional effects between high and moderate compliers indicates that there may be an implementation 'sweet spot' such that more is not always better [7]. For instance, high compliance to activity structure and timing (i.e., adherence) may require teachers to go against their professional judgement to adapt lesson delivery to their unique class context [91]. Teachers who prioritised adherence to activity timing over student-led discussion thought that Passport lessons were less effective [92]. Indeed, such a top-down prescriptive approach to teaching is criticised as a 'democratic deficit' in education [93]. Moderate compliers that adhere comparatively less to a top-down delivery model, and in turn remain responsive to student-led discussion, may more successfully foster an environment conducive to SEL [91].

The finding that high quality delivery is associated with negative effects for peer social support is harder to explain; Passport may inadvertently nurture emotional and social awareness without providing students with skills to seek support or to managing complex social situations. Findings from the process evaluation support this hypothesis, as some teachers described adapting programme content to their class (an indicator of quality) which resulted in inadvertently removing essential activities that award help-seeking (e.g., the Help Thermometer) and the identification of coping skills (the Dragon's Path); indeed, some students struggled to apply strategies in practice, particularly in the context of persistent bullying [94]. It is possible that greater exposure to lesson content, without sufficient structural support to practice skills, prevented development of targeted skills required to navigate social challenges.

This theory can be further interrogated in the context of an unexpected trend towards poorer scores for high attending classrooms (i.e., reach). The effectiveness of Passport may be compromised in large classroom settings if students have fewer opportunities to engage meaningfully in discussions or practice-based activities. Teacher interviews suggest that managing activities became increasingly challenging with over 30 students, particularly as Passport is student-led and centres around peer group discussions [92]. Passport was initially designed and piloted in schools with an average class size of 20 pupils [11] yet the programme is now delivered in English classrooms [67] which have on average 26 pupils [95], as reflected in the current trial's sample. This 30% increase in class size may disrupt the programme's core components and prevent students from deriving the intended benefits from guided practice elements (e.g., role play, games) that are essential to building confidence and competency to navigate such social challenges successfully.

The unexpected finding that high reach is paradoxically related to poorer outcomes (d = .21 −.32), compared to moderate reach, could be further explained by drawing upon the economic theory of opportunity cost, or the value of the next-best alternative that must be forgone when making a choice [96]. Delivering a universal programme like Passport requires schools to make resource allocation decisions within finite timetables and staffing constraints [97] – in other words, Passport must replace another lesson [98]. Teachers who report high attendance may choose to deliver a universal programme like Passport as an alternative to targeted support, which is more costly [99]; in contrast, moderate attendance may reflect a strategic approach, whereby an intervention like Passport is delivered when a portion of students who require targeted support leave the classroom to engage in those activities. Although critical to informing economic evaluations of school-based interventions [100], it's unclear how opportunity cost plays out in practice [101]. Further qualitative interviews were

conducted with school staff to understand how perceptions of an intervention's value and cost informs decisions to adopt and implement a universal intervention like Passport [92].

Finally, this trend towards poorer outcomes among high compliers may be explained by a complementary phenomenon of 'response shift' [102], whereby an intervention that aims to improve student outcomes (e.g., emotional literacy) improves measurement accuracy post-intervention. We recommend that future studies consider using differential item functioning analyses to examine whether control and intervention participants interpret measure items differently [103]. While previous research found no difference in teachers' interpretation of bullying behaviours following delivery of the PATHS programme [104], to the author's knowledge, the possibility of response shift has not yet been explored among children participating in school-based SEL interventions.

### Considering measurement error

As per our trial protocol, implementation data was collected using a cross-sectional teacher-report survey to minimise burden and mitigate potential attrition compared to, for example, teacher observations [43]. This approach required teachers to provide a global rating across the 18-week intervention period, rather than session-specific assessments. This methodological constraint, though necessary, may have resulted in several limitations.

First, this approach limits identification and examination of within-programme fluctuation across implementation dimensions [7]. Second, the retrospective design introduces potential for recall bias, particularly recency effects, whereby scores may have been disproportionally influenced by the final 'Celebration' session (which was thoroughly enjoyed by staff and students) [94]. Third, self-report implementation is subject to social desirability bias, whereby responses are inflated to align with perceived expectations or professional standards [105,106]. Accordingly, the consistency of results across conceptually distinct implementation dimensions may result from reliance on single-reporter implementation data, inflating intercorrelations among dimensions.

The optimal approach to assessing implementation in school-based interventions is third-party observations [9,107] with an increasing number of tools available to researchers [108,109]. Observations offer value in terms of objectivity and rigour, but can be resource intensive and burdensome on school staff [7] and are not immune to bias (e.g., observer effect) [107]. Additionally, the inter-rater reliability between teacher and observer fidelity measures tends to be low [105,110]. It is therefore essential to consider approaches to optimising self-report measures, such as assessing student responsiveness using student-report surveys (rather than teacher-report) [9] and involving participants in survey design and development [111]. One such approach is ecological momentary assessment (EMA) to capture participant implementation behaviours (e.g., student responsiveness, teacher quality) during programme delivery. EMA capture variation in behaviours and attitudes across time and contexts, providing insight into whether and how responses vary between and across participants groups, contexts and programme sessions [112].

These suggestions align with calls to embed implementation considerations during programme design rather than an "added and separate effort" [113], and are important given evidence that researchers and practitioners may conceptualise implementation constructs differently [e.g., fidelity; 114]. This was observed in the current trial when teachers simultaneously reported high fidelity (i.e., adherence to lesson learning objectives, activity sequence and structure, and programme manual for core activities), while detailing lesson adaptations, modifications, and omissions [94]. This conceptual distinction could be explained by whether teachers are considering and reporting pedagogical adherence (*how* it is taught, e.g., role play) or practice-based adherence (*what* is taught, e.g., identifying emotions) [115]. Teachers draw upon their pedagogical knowledge to adapt an SEL intervention [116] and may therefore still report high fidelity if they believe they are adhering to the programme *ethos* despite pedagogical deviations [117]. The construct validity of implementation measures is rarely assessed or reported in school-based SEL research, raising concerns about the accuracy with which current tools capture the intended construct [9]. For example, the current measure of quality asked teachers if they felt able to engage

pupils and well-prepared to teach lessons; one could argue that these items are indicators of programme design rather than quality of teacher delivery.

We urge researchers to carefully consider best practice when assessing implementation variability in school-based trials [106]. Ideally, implementation data is collected from multiple informants (e.g., student responsiveness self-reported by students) and at multiple timepoints. Moving forwards, researchers using multi-method multi-informant approaches (e.g., event-contingent EMA) would benefit from extending the multiverse framework to explore whether seemingly homogenous results in a dataset are the result of measurement issues outlined above or a well-delivered intervention [118]. Using a study within a trial (SWAT) design can provide valuable insights for advancing school-based process evaluations [119], such as the validity of approaches to collecting implementation data (e.g., weekly, cross-sectional). Our Implementation Quality Appraisal Checklist provides tiered guidance to improve research practice, which can be useful when managing resource constraints [9].

## Moving from individual parts to a whole picture

The multiverse reveals variability in results depending on how we define and model intervention compliance, which manifested in entropy values, precision of confidence intervals and the strength and direction of intervention effects. Despite this variability, clear patterns emerged across the multiverse: certain outcomes (e.g., victimisation) produced stronger entropy values than others (e.g., loneliness) and moderate compliance was more often associated with positive outcomes compared to high compliance.

When considering the results of the multiverse as a whole, it instils confidence in findings that were not variable across analytic universes: all but one of the 27 CACE models were not statistically significant, indicating that the CACE effects are largely consistent with the ITT results and Passport was no better than usual practice in improving children's relational outcomes, however it is delivered. This further supports the hypothesis that null intervention effects are not a result of implementation failure but can be explained by low programme differentiation [64], as approximately 60% of teachers reported implementing similar universal SEL programmes that overlapped substantially with Passport content. This finding is not unexpected given that schools are encouraged to prioritise SEL due to robust evidence of its efficacy [120].

Our findings reinforce the importance of considering implementation behaviours as distinct constructs and thus shaped by different determinants [6,89]. Future research would benefit from further empirical work examining student, teacher and classroom characteristics that predict implementation behaviours [121]. This would not only result in more robust modelling but have important practical implications; identifying individuals or contexts at greater risk of non-compliance can inform targeted allocation of resources to promote equitable implementation. For researchers utilising the current paper to guide implementation and process evaluation, we recommend using theory and evidence to prospectively identify appropriate predictors to enhance model robustness, for example, using qualitative and quantitative evidence of factors influencing implementation behaviours from systematic reviews. Additionally, involving teachers at participating schools via participatory co-design could inform the final selection of compliance predictors ensuring contextual relevancy. These recommendations have value beyond CACE; while its critics propose using a propensity score approach as a more robust alternative [52], this approach similarly relies on good covariates to predict compliance [45].

Furthermore, the findings from the multiverse challenge the assumption that more is always better when delivering a universal SEL programme like Passport. Considering that time is a major barrier to programme delivery in schools [90], our results support the continued calls to empirically establish intervention core components. This could be achieved using a factorial trial design [115,122] or bottom-up network approaches, whereby variation in intervention outcomes are mapped onto the associated active ingredients delivered that week [123]. This suggestion may draw scepticisms due to the subsequent burden on intervention participants, particularly amidst continued challenges with recruiting schools to trials [124,125]. Subsequently, secondary data analysis may be an appropriate solution [e.g., 126]. Such insights could

refine intervention models, removing redundant activities to produce a streamlined intervention (i.e., how much is good enough) [7,127].

Finally, we are encouraged to practice caution when null and/or negative intervention effects juxtapose participants' experiences of a programme's effectiveness [128]. The qualitative strand of the Passport trial found that students thoroughly enjoyed Passport and many teachers expressed an interest in continuing programme delivery for future year groups [94]. The programme manual provided teachers with clear structure and guidance for teaching SEL, and supported strong implementation, as indicated by the negatively skewed implementation data. This aligns with teacher preferences for clear, structured and time-efficient SEL curricula [129]. It is therefore possible that a manualised SEL programme like Passport, with minimal planning and preparation time, provides additional benefits for *teachers*, rather than students, when compared to the usual curriculum, given that educators' time is one of the most valuable resources available to schools [99]. However, our findings highlight a critical gap in our understanding of how schools navigate the opportunity cost, or trade-off, when deciding whether to adopt and deliver a universal intervention like Passport. These decision-making processes that guide resource allocation are poorly understood, particularly in the context of competing demands and resource constraints [97,98,129]. Further research will explore what drives school's programme preferences, and how this underpins resource allocation, which is essential to understanding and predicting programme adoption, implementation and sustainability patterns [92].

## Conclusions

Evaluating implementation variability in school-based trials is not easy, but it is essential [8]. This paper demonstrates the use of multiverse analysis to systematically assess, and transparently report, how arbitrary decisions in our CACE model can influence results and subsequent conclusions. Importantly, it highlights how reliance on dosage to define compliance in education may obscure critical information about an intervention's effectiveness, particularly if an intervention has few sessions. By applying a multiverse framework to CACE, this paper employs a new approach to address an old problem; specifically, it provides an analytic framework for modelling implementation variability for school-based trials. In doing so, we move from a fragmented view of the implementation-outcomes relationship, towards a more coherent understanding of the complexity of examining complex interventions in real-world settings, seeing not just the parts, but the whole.

## Supporting information

**S1 File. Outcome surveys.**
(DOCX)

**S2 File. Multiverse decision making.**
(DOCX)

**S3 File. Psychometric properties.**
(DOCX)

**S4 File. Model results.**
(XLSX)

## Acknowledgments

We extend our sincerest gratitude to schools, teachers and students who participated in the Passport to Success trial.

## Author contributions

**Conceptualization:** Annie O'Brien, Margarita Panayiotou.

**Data curation:** Joao Santos.

**Formal analysis:** Annie O'Brien.

**Funding acquisition:** Neil Humphrey.

**Investigation:** Annie O'Brien.

**Methodology:** Annie O'Brien, Margarita Panayiotou.

**Supervision:** Joao Santos, Neil Humphrey, Margarita Panayiotou.

**Visualization:** Annie O'Brien.

**Writing – original draft:** Annie O'Brien.

**Writing – review & editing:** Annie O'Brien, Joao Santos, Neil Humphrey, Margarita Panayiotou.

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
