## [Editor Report · Decision Letter 0]

25 Nov 2025

PONE-D-25-61187

CACE closed: A multiverse examination of the influence of implementation variability on student outcomes in a randomised controlled trial of a universal, school-based social-emotional learning intervention

PLOS ONE

Dear Dr. O'Brien,

Thank you for submitting your manuscript to PLOS ONE. After careful consideration, we have decided that your manuscript does not meet our criteria for publication and must therefore be rejected.

Specifically:

The paper proposes a methodological approach to account for implementation variability in a cluster randomised trial. It shows that different assumptions about implementation levels yield different meassures of effect size. Sensitivity analysis serves as a useful tool for examining variations after an intention-to-treat analysis. However, allowing researchers to define multiple compliance thresholds could lead to spurious results and reduce study power by testing multiple hypotheses. Although formal approaches to assess implementation variability are valuable, the manuscript risks creating misunderstandings about sensitivity analysis and its role in research.

I am sorry that we cannot be more positive on this occasion, but hope that you appreciate the reasons for this decision.

Kind regards,

Ivan Sarmiento

Academic Editor

PLOS ONE

- - - - -

---

## [Author Response · Author response to Decision Letter 1]

16 Dec 2025

Dear Prof Sarmiento,

I am writing to formally appeal the decision to reject our manuscript, titled CACE closed: A multiverse examination of the influence of implementation variability on student outcomes in a randomised controlled trial of a universal, school-based social-emotional learning intervention (PONE-D-25-61187).

My co-authors and I sincerely thank you for the time and effort dedicated to reviewing our submission. We are disappointed by the decision not to send our paper for external peer review, which we believe stems from a misunderstanding of our analytic approach. We wish to clarify that our use of sensitivity analysis with pre-specified compliance thresholds is distinct from post-hoc data exploration and aligns with recommended practices in educational intervention research and recent publications in PLOS ONE. Below, we provide a point-by-point response detailing our rationale for reconsidering the decision to reject this manuscript prior to external peer review.

Comment 1: Allowing researchers to define multiple compliance thresholds could lead to spurious results and reduce study power by testing multiple hypotheses.

Response: We appreciate this methodological concern but believe it stems from a misunderstanding of our analytic approach. Our study adopts a pre-registered multiverse framework (https://osf.io/s5pmw/overview) specifically designed to address the problem of spurious results arising from researcher degrees of freedom in defining and modelling compliance.

First, we address the concern about defining two compliance thresholds.

CACE is limited by its reliance on a binary marker of compliance because defining 'full' compliance of a prevention intervention is significantly challenging. CACE is dependent on the exclusion restriction assumption (non-compliers derive no benefit from the intervention) which is difficult to validate in the absence of theory-informed compliance thresholds (Education Endowment Foundation, 2018) which are largely untested and unknown in psychological and educational interventions (Durlak, 2010). Accordingly, Peugh and colleagues (2017) recommend: "a series of additional sensitivity analyses could be conducted using several binary compliance indicator (“U”) variables defined at multiple cut-points (e.g., 50%, 70%, 90%, 95%) and observing the change in the CACE treatment effect estimate” (p.23).

We adopt this advice, but rather than making a single arbitrary choice when defining ‘full’ compliance and subsequently presenting misleading results, we pre-registered a sensitivity analysis examining two compliance thresholds (50th and 75th percentile). This choice follows established practice in educational research (e.g., Ashworth et al., 2020; Berg et al., 2017; Bradshaw et al., 2020; Humphrey & Panayiotou, 2022; Humphrey et al., 2021; Panayiotou et al., 2020). This decision was also pragmatic, as each CACE model requires substantial computational time (>24 hours) and it will facilitate comparison with other intervention studies using identical thresholds, such as the ones cited above.

Our pre-registration details the above rationale, which we hope instils in the Editor our recognition of the importance of this discussion. This is signposted on page 17 (lines 405, 11 - 415) of our manuscript, detailed in a supplementary file (File S2) and reported in our preregistration (https://osf.io/s5pmw/overview).

Second, we address the concern about reducing study power by testing multiple hypotheses briefly here, and in more detail below.

A correction for multiplicity was not applied in the current study as the multiverse is exploratory: the goal is not to correcting for multiplicity (i.e., alpha adjustment) but to increase transparency by visualising the distribution of outcomes. This is reported in the manuscript on page 19 lines 433-436.

Action taken: We have revised the introduction (page 8 lines 192-202) and the methods (page 17 lines 405, 11 - 415). These sections now discuss the challenge in defining full compliance in prevention interventions, report the recommendation to carry out sensitivity analyses and cites empirical research demonstrating this practice in education. Should this manuscript be reconsidered, and further detail be required, we are happy to be guided on this.

Comment 2: Although formal approaches to assess implementation variability are valuable, the manuscript risks creating misunderstandings about sensitivity analysis and its role in research.

Response: We take this concern seriously and want to ensure that our work contributes to methodological clarity rather than obscures it.

PLOS ONE has championed the use of multiverse analysis to assess robustness across analytical decisions (e.g., Claesen et al., 2023; Dürlinger et al., 2022) and highlighted how such approaches remain underutilised (e.g., Böschen, 2023). Our study extends the application of this framework to school-based intervention research to examine whether conclusions remain stable when there are a range of reasonable ways to define compliance. We report on page 22 (lines 519-522) that the use of a multiverse framework is not to discover which model conditions result in desirable outcomes (i.e., cherry picking), but whether results are robust to these variations.

Our pre-registered multiverse framework aims to address, rather than exacerbate, concerns about multiple testing by transparently reporting pragmatic and plausible specifications. This differs from post-hoc specification searching, selective reporting, or testing multiple hypothesis without adjustment. A multiverse analysis framework does not inherently require an alpha adjustment for multiplicity because its primary goal is to assess the robustness of a finding across all plausible analytical paths, rather than to make a single, definitive confirmatory claim based on a chosen one (Clayson, 2024). The exploratory nature of these analyses are noted on page 19, lines 424-6 of our manuscript.

We recognise that our manuscript may benefit from clearer articulation of when such analytic approaches are appropriate and the inferential limitations. Should our manuscript be reconsidered, we would welcome specific guidance on where our presentation risks creating misunderstandings and would revise accordingly.

Thank you for your time and consideration of our request. We look forward to hearing from you.

Best wishes,

Annie O’Brien, on behalf of the authorship team

---

## [Decision Letter · Decision Letter 1]

27 Feb 2026

PONE-D-25-61187R1CACE closed: A multiverse examination of the influence of implementation variability on student outcomes in a randomised controlled trial of a universal, school-based social-emotional learning interventionPLOS One

Dear Dr. O'Brien,

Thank you for submitting your manuscript to PLOS ONE. After careful consideration, we feel that it has merit but does not fully meet PLOS ONE’s publication criteria as it currently stands. Therefore, we invite you to submit a revised version of the manuscript that addresses the points raised during the review process. Reviewers' 2 and 3 are generally on board with the paper as it stands. This is a good signal, and it aligns with Reviewer 1's emphasis that the manuscript is of scientific merit and worth of publication in PLOS One. At the same time Reviewer 1 identifies key areas that make the paper A) difficult to follow and understand its contribution, and B) possibly overstated in what it claims. I encourage the author to revise the paper responding carefully to the comments by Reviewer 1 and a point raised by Reviewer 3 about the role of the R2 here. Reviewer 1 pointed out that the use of 'researcher degrees of freedom' is confused and confusing. These are the theoretical range of choices a researcher has in specifying a model or in a workflow, it is not fully known. Moreover, I agree with R1 that this paper introduces a solution to problems with analyzing the data. It is a paper that recommends a multiverse method, but it needs to be made clearer at the beginning that a demonstration of this method is done within a specific study which is a case study of similar studies that suffer from this/these problems. I look forward to a revised version of this manuscript.

We look forward to receiving your revised manuscript.

Kind regards,

Nate Breznau

Academic Editor

PLOS One

Journal Requirements:

2.Please note that funding information should not appear in any section or other areas of your manuscript. We will only publish funding information present in the Funding Statement section of the online submission form. Please remove any funding-related text from the manuscript.

3.Thank you for stating the following financial disclosure:

"This work is part of the first author’s PhD research, funded by the Kavli Trust (https://kavlifondet.no/en/), grant no. Kavli2021-0000000019."

4.When completing the data availability statement of the submission form, you indicated that you will make your data available on acceptance. We strongly recommend all authors decide on a data sharing plan before acceptance, as the process can be lengthy and hold up publication timelines. Please note that, though access restrictions are acceptable now, your entire data will need to be made freely accessible if your manuscript is accepted for publication. This policy applies to all data except where public deposition would breach compliance with the protocol approved by your research ethics board. If you are unable to adhere to our open data policy, please kindly revise your statement to explain your reasoning and we will seek the editor's input on an exemption. Please be assured that, once you have provided your new statement, the assessment of your exemption will not hold up the peer review process.

6. Please include a separate legend for each figure in your manuscript.

Reviewers' comments:

Reviewer's Responses to Questions

**Comments to the Author**

1. If the authors have adequately addressed your comments raised in a previous round of review and you feel that this manuscript is now acceptable for publication, you may indicate that here to bypass the “Comments to the Author” section, enter your conflict of interest statement in the “Confidential to Editor” section, and submit your "Accept" recommendation.

Reviewer #1: (No Response)

Reviewer #2: All comments have been addressed

Reviewer #3: All comments have been addressed

2. Is the manuscript technically sound, and do the data support the conclusions?

Reviewer #1: Yes

Reviewer #2: Yes

Reviewer #3: Yes

3. Has the statistical analysis been performed appropriately and rigorously? 

Reviewer #1: Yes

Reviewer #2: Yes

Reviewer #3: No

4. Have the authors made all data underlying the findings in their manuscript fully available?

Reviewer #1: Yes

Reviewer #2: Yes

Reviewer #3: Yes

5. Is the manuscript presented in an intelligible fashion and written in standard English?

Reviewer #1: Yes

Reviewer #2: Yes

Reviewer #3: Yes

6. Review Comments to the Author

Reviewer #1: In their manuscript “CACE closed: A multiverse examination of the influence of implementation variability on student outcomes in a randomised controlled trial of a universal, school-based social-emotional learning intervention”, the authors report a preregistered secondary analysis of a large randomised control trial evaluating a school-based social-emotional learning intervention (Passport). Notably, the authors apply a preregistered multiverse analysis to complier average causal effect modelling to report whether results on relational outcomes are robust to defensible analytical flexibility regarding how compliance is defined and thresholded. Across their multiverse of 27 pipelines, the authors report consistently null effects, with one isolated statistically significant negative effect for peer support under a high-reach specification. The central conclusion is that substantive inferences about intervention effectiveness are sensitive to researcher degrees of freedom in CACE modelling, and that a multiverse approach offers a transparent way to report robustness in implementation science.

I would like to begin by expressing support for the application of a multiverse framework. A growing body of work across scientific disciplines has demonstrated that defensible analytic flexibility can influence the reported results, and selective reporting of a single analytic pipeline risks overstating confidence in conclusions. I believe that the multiverse analysis approach offers a systematic and comprehensive framework to report the robustness of results across researcher degrees of freedom. It is encouraging to see this approach applied to compliance-adjusted casual inference in educational research, and the theoretical motivation for this application is well-grounded in the manuscript.

The manuscript has several additional strengths. The analyses are preregistered, the authors appropriately emphasise robustness, entropy and uncertainty rather than statistical significance alone, and the discussion is generally cautious and methodologically reflective.

However, I have identified several major and minor issues that, if addressed, would strengthen the manuscript prior to publication. My expertise is particularly well aligned with the evaluation of the multiverse analysis and its interpretation.

Major issues

First, the introduction would benefit greatly from restructuring. Currently, it begins with an explanation of multiverse analysis, which is followed by the rationale for its application to the substantive topic of the manuscript. I believe clarity would be improved by first introducing the substantive problem, and then presenting multiverse analysis as a methodological solution for transparently reporting uncertainty. As written, the opening gives the impression that the manuscript primarily advances multiverse analysis as a method, rather than using it as a tool to address a specific substantive question.

Second, the manuscript occasionally overstates what multiverse analysis achieves. For example, the authors suggest that multiverse analysis can be used to ‘limit researcher degrees of freedom’. In its current application, the multiverse analysis does not identify superior pipelines from those computed. Rather, it documents how results vary across defensible choices. In this sense, multiverse analysis reports uncertainty arising from researcher degrees of freedom rather than constraining them. The role of multiverse analysis as a means to report uncertainty is explained within this paper by Hoffmann and colleagues:

Hoffmann, S., Schönbrodt, F., Elsas, R., Wilson, R., Strasser, U., & Boulesteix, A. L. (2021). The multiplicity of analysis strategies jeopardizes replicability: lessons learned across disciplines. Royal Society Open Science, 8(4), 201925. https://doi.org/10.1098/rsos.201925

And a recent community opinion on what multiverse analysis can and cannot address is presented in a preprint, along with guidelines for its implementation:

Short, C. A., Breznau, N., Bruntsch, M., Burkhardt, M., Busch, N., Cesnaite, E., ... & Hildebrandt, A. (2025). Multi-curious: A multi-disciplinary guide to a multiverse analysis. MetaArXiV. https://doi.org/10.31222/osf.io/4yzeh_v2

In lines 221-222, the authors state that ‘researcher degrees of freedom still limit this method’. I wonder whether it would be more accurate to say that unreported researcher degrees of freedom is rather limiting replicability and, when unreported, leads to uncertainty. This my be more conceptually precise than arguing that researcher degrees of freedom limits the method.

Third, as many readers may not yet be familiar with the multiverse analysis framework, the manuscript would benefit from a concise, explicit statement, ideally in the methods or results section, clarifying the claims that can and cannot be made based on the multiverse analysis presented (e.g., descriptive robustness report rather than identifying superior analytical decisions). This may prevent the assumption that the pipeline producing a significant effect is methodologically superior to the other pipelines, which cannot be deduced from the present results.

Fourth, greater transparency is needed regarding how the set of analytic options included in the multiverse was identified. The authors cite Del Giudice and Gangestad (2021) as guiding the construction of the multiverse, but this framework primarily concerns equivalence among pipelines rather than procedures for identifying candidate options in the first place. While Supplementary File 2 provides additional detail, some justifications (for example, excluding options because ‘both specifications are known to result in a different outcome’) are unclear. Different outcomes of which metric, and why this constitutes grounds for exclusion? Multiverse analyses have previously identified analytic options through approaches such as systematic literature review, crowdsourcing, or expert consensus. The authors should clearly report where the pool of candidate options originated, how inclusion and exclusion decisions were made, and what procedure led to the final set of pipelines. Adopting the structured guidance proposed by Short et al. (2025) may help strengthen transparency, allow readers to evaluate potential bias, and aid reproducibility.

Fifth, there is considerable repetition throughout the manuscript, including within individual paragraphs. For example, in the Statistical Analysis section (beginning on line 420), the justification for using two-level models with SE adjustment is repeated across successive sentences. In particular, the rationale given in the third and fifth sentences of this paragraph could be combined to convey the same information more concisely. Similar repetition occurs elsewhere in the manuscript, and reducing it would substantially improve clarity and readability without loss of content.

Minor issues:

Sixth, the multiverse plots may be more visually informative if presented in a specification curve style format. A visual summary of the pipelines would also be helpful, to quickly deduce which options and combinations of options were included in the multiverse analysis. This could be in the format of a sankey plot.

See specification curve plot here: Simonsohn, U., Simmons, J. P., & Nelson, L. D. (2020). Specification curve analysis. Nature Human Behaviour, 4(11), 1208-1214. https://doi.org/10.1038/s41562-020-0912-z

Seventh, I am not sure that the elephant analogy adds clarity or scientific meaning (e.g., referring to entropy values as ‘the elephant’s trunk’, precision and confidence intervals as 'the elephant's tusk’, direction and effect sizes as ‘the elephant’s tail’ etc.). For readers new to multiverse analysis, I think this analogy may add to the cognitive load for readers unfamiliar with multiverse analysis and potentially cause confusion. I understand this is perhaps personal preference, but I would prefer direct explanations.

Finally, the authors should complete a final proof reading of the manuscript to correct mistakes such as un-paired brackets in the text (e.g., line 156) and double commas (e.g., line 208).

Overall, I believe this is a methodologically rigourous, well-motivated and transparent study that fits within PLOS ONE’s publication criteria. Addressing the points above would substantially strengthen clarity and guard against misinterpretation.

Reviewer #2: The initial criticism regarding "spurious results" and "reduced study power" appears to conflate traditional confirmatory hypothesis testing with Multiverse Analysis. The manuscript employs a pre-registered multiverse framework (OSF), which fundamentally mitigates the risks of p-hacking or post-hoc data dredging. By testing 10 pre-specified compliance (CACE) models across distinct theoretical dimensions (such as fidelity, dosage, and reach), the authors are not "cherry-picking" a p-value; rather, they are mapping the robustness and stability of the intervention effect against the inherent "researcher degrees of freedom" that often plague school-based trials.

From a causal inference perspective, defining a "complier" in a real-world educational setting is a notorious methodological challenge. The authors' decision to apply multiple thresholds (50th and 75th percentiles) aligns with cutting-edge recommendations from scholars like Peugh et al. (2017) and institutions such as the Education Endowment Foundation. This transparency is essential for validating the exclusion restriction assumption of the CACE model, ensuring that the findings regarding the "Passport" intervention are not merely artifacts of a single, arbitrary statistical cutoff.

Ultimately, this work represents an exemplary application of rigorous, open science that fits perfectly within the scope of PLOS ONE. It provides a valuable contribution to implementation science by demonstrating how varying compliance metrics can shift, or confirm, an effect.

Reviewer #3: 1. It is noted that the statistical analysis of the tables attached in Supplementary File 2 was not adequately addressed. For example, the significance of the R2 value, its significance, and the justification for its decrease in some cases. Likewise, the analysis of what was stated in Supplementary File 3 suffers from a lack of depth in expansion and statistical significance.

2. It was suggested that a 50% threshold would suffice for the compliance standard.

3. I believe that analyzing sensitivity is appropriate for the study.

7. PLOS authors have the option to publish the peer review history of their article (what does this mean?). If published, this will include your full peer review and any attached files.

Reviewer #1: **Yes:**Cassie Short

Reviewer #2: No

Reviewer #3: No

---

## [Author Response · Author response to Decision Letter 2]

1 Apr 2026

We thank the reviewers for their careful reading of the manuscript and for their constructive and thoughtful comments. We have responded to each comment in turn below and have revised the manuscript accordingly where appropriate. We believe that these revisions have strengthened the clarity, rigour, and contribution of the paper, and we are grateful for the opportunity to address the reviewers’ feedback.

Reviewer #1

In their manuscript “CACE closed: A multiverse examination of the influence of implementation variability on student outcomes in a randomised controlled trial of a universal, school-based social-emotional learning intervention”, the authors report a preregistered secondary analysis of a large randomised control trial evaluating a school-based social-emotional learning intervention (Passport). Notably, the authors apply a preregistered multiverse analysis to complier average causal effect modelling to report whether results on relational outcomes are robust to defensible analytical flexibility regarding how compliance is defined and thresholded. Across their multiverse of 27 pipelines, the authors report consistently null effects, with one isolated statistically significant negative effect for peer support under a high-reach specification. The central conclusion is that substantive inferences about intervention effectiveness are sensitive to researcher degrees of freedom in CACE modelling, and that a multiverse approach offers a transparent way to report robustness in implementation science.

I would like to begin by expressing support for the application of a multiverse framework. A growing body of work across scientific disciplines has demonstrated that defensible analytic flexibility can influence the reported results, and selective reporting of a single analytic pipeline risks overstating confidence in conclusions. I believe that the multiverse analysis approach offers a systematic and comprehensive framework to report the robustness of results across researcher degrees of freedom. It is encouraging to see this approach applied to compliance-adjusted casual inference in educational research, and the theoretical motivation for this application is well-grounded in the manuscript.

The manuscript has several additional strengths. The analyses are preregistered, the authors appropriately emphasise robustness, entropy and uncertainty rather than statistical significance alone, and the discussion is generally cautious and methodologically reflective.

However, I have identified several major and minor issues that, if addressed, would strengthen the manuscript prior to publication. My expertise is particularly well aligned with the evaluation of the multiverse analysis and its interpretation.

Response: We wish to express our sincere gratitude for your time, expertise, and words of support for this manuscript. We have taken feedback on board and believe it has significantly strengthened this manuscript and for that we are most thankful.

Major issue 1: First, the introduction would benefit greatly from restructuring. Currently, it begins with an explanation of multiverse analysis, which is followed by the rationale for its application to the substantive topic of the manuscript. I believe clarity would be improved by first introducing the substantive problem, and then presenting multiverse analysis as a methodological solution for transparently reporting uncertainty. As written, the opening gives the impression that the manuscript primarily advances multiverse analysis as a method, rather than using it as a tool to address a specific substantive question.

Thank you for this helpful comment – upon revisiting the manuscript, we agree that restructuring the introduction would improve the clarity of our aims as well as the readability of our manuscript.

Response: We have restructured the introduction as requested.

Second, the manuscript occasionally overstates what multiverse analysis achieves. For example, the authors suggest that multiverse analysis can be used to ‘limit researcher degrees of freedom’. In its current application, the multiverse analysis does not identify superior pipelines from those computed. Rather, it documents how results vary across defensible choices. In this sense, multiverse analysis reports uncertainty arising from researcher degrees of freedom rather than constraining them. The role of multiverse analysis as a means to report uncertainty is explained within this paper by Hoffmann and colleagues:

Hoffmann, S., Schönbrodt, F., Elsas, R., Wilson, R., Strasser, U., & Boulesteix, A. L. (2021). The multiplicity of analysis strategies jeopardizes replicability: lessons learned across disciplines. Royal Society Open Science, 8(4), 201925. https://doi.org/10.1098/rsos.201925 [doi.org]

And a recent community opinion on what multiverse analysis can and cannot address is presented in a preprint, along with guidelines for its implementation:

Short, C. A., Breznau, N., Bruntsch, M., Burkhardt, M., Busch, N., Cesnaite, E., ... & Hildebrandt, A. (2025). Multi-curious: A multi-disciplinary guide to a multiverse analysis. MetaArXiV. https://doi.org/10.31222/osf.io/4yzeh_v2 [doi.org]

In lines 221-222, the authors state that ‘researcher degrees of freedom still limit this method’. I wonder whether it would be more accurate to say that unreported researcher degrees of freedom is rather limiting replicability and, when unreported, leads to uncertainty. This my be more conceptually precise than arguing that researcher degrees of freedom limits the method.

Response: Thank you for sharing this paper and preprint – they have both been enjoyable reads and are thought provoking. We understand the concern raised about how current phrasing overstates what it is that the multiverse can and cannot achieve. We have clarified our reporting of the aims and outcomes of the multiverse analysis. For examples, please see:

- Page 2 lines 36-48

- Page 10 lines 234-250

- Page 11 lines 270-271

- Page 33 lines 809-812

Third, as many readers may not yet be familiar with the multiverse analysis framework, the manuscript would benefit from a concise, explicit statement, ideally in the methods or results section, clarifying the claims that can and cannot be made based on the multiverse analysis presented (e.g., descriptive robustness report rather than identifying superior analytical decisions). This may prevent the assumption that the pipeline producing a significant effect is methodologically superior to the other pipelines, which cannot be deduced from the present results.

Response: Thank you for raising this - we have expanded upon our note of caution in the discussion section (page 25 lines 566-572): We recommend caution when interpreting the results of the multiverse, which does not identify superior analytic decisions based on, for example, statistical significance, rather it provides a descriptive report of the robustness and replicability of results [12]. In other words, its purpose is not to discover which model conditions result in desirable outcomes (i.e., cherry picking), but transparently report whether results are robust to these variations.

Fourth, greater transparency is needed regarding how the set of analytic options included in the multiverse was identified. The authors cite Del Giudice and Gangestad (2021) as guiding the construction of the multiverse, but this framework primarily concerns equivalence among pipelines rather than procedures for identifying candidate options in the first place. While Supplementary File 2 provides additional detail, some justifications (for example, excluding options because ‘both specifications are known to result in a different outcome’) are unclear. Different outcomes of which metric, and why this constitutes grounds for exclusion? Multiverse analyses have previously identified analytic options through approaches such as systematic literature review, crowdsourcing, or expert consensus. The authors should clearly report where the pool of candidate options originated, how inclusion and exclusion decisions were made, and what procedure led to the final set of pipelines. Adopting the structured guidance proposed by Short et al. (2025) may help strengthen transparency, allow readers to evaluate potential bias, and aid reproducibility.

Response: We have revised File S2. This file now contains detailed decisions about how we identified candidate options for the multiverse, the available options, and the decisions made. This now includes explicit reference to our systematic review and crowdsourcing approach.

Fifth, there is considerable repetition throughout the manuscript, including within individual paragraphs. For example, in the Statistical Analysis section (beginning on line 420), the justification for using two-level models with SE adjustment is repeated across successive sentences. In particular, the rationale given in the third and fifth sentences of this paragraph could be combined to convey the same information more concisely. Similar repetition occurs elsewhere in the manuscript, and reducing it would substantially improve clarity and readability without loss of content.

Response: We have amended this section as requested. We have also reviewed the manuscript and made changes throughout to reduce repetition. For example:

- Page 8 lines 176-186

- Page 12 lines 264-281

- Page 17-18 lines 384-388

- Page 18 lines 392-403

Minor issues: Sixth, the multiverse plots may be more visually informative if presented in a specification curve style format. A visual summary of the pipelines would also be helpful, to quickly deduce which options and combinations of options were included in the multiverse analysis. This could be in the format of a sankey plot.

See specification curve plot here: Simonsohn, U., Simmons, J. P., & Nelson, L. D. (2020). Specification curve analysis. Nature Human Behaviour, 4(11), 1208-1214. https://doi.org/10.1038/s41562-020-0912-z [doi.org]

Thank you for these suggestions.

Response: We believe that an advantage of the existing figures is that they are simple, intuitive and accessible to those unfamiliar with multiverse analysis and specification curve analysis. However, upon review of our figures we do think that they could be improved for readability. We therefore have reduced the length of the x-axis for each figure (see updated Figures 2-4).

We agree that it would be helpful to include a visual illustration of the multiverse pipelines. As Sankey plots are commonly used for longitudinal data, we thought it best to use a tree diagram to visually illustrate this information. We have depicted this in a new figure (Figure 1).

Seventh, I am not sure that the elephant analogy adds clarity or scientific meaning (e.g., referring to entropy values as ‘the elephant’s trunk’, precision and confidence intervals as 'the elephant's tusk’, direction and effect sizes as ‘the elephant’s tail’ etc.). For readers new to multiverse analysis, I think this analogy may add to the cognitive load for readers unfamiliar with multiverse analysis and potentially cause confusion. I understand this is perhaps personal preference, but I would prefer direct explanations.

Response: This analogy has been removed from the manuscript.

Finally, the authors should complete a final proof reading of the manuscript to correct mistakes such as un-paired brackets in the text (e.g., line 156) and double commas (e.g., line 208).

Response: Thank you for spotting these punctuation mistakes which have now been corrected. We have also reviewed the manuscript and corrected remaining grammatical errors.

Overall, I believe this is a methodologically rigourous, well-motivated and transparent study that fits within PLOS ONE’s publication criteria. Addressing the points above would substantially strengthen clarity and guard against misinterpretation.

Response: Thank you for your constructive and supportive feedback on our manuscript. We have implemented the requested changes and believe that the manuscript has been significantly strengthened as a result.

Reviewer 2

The initial criticism regarding "spurious results" and "reduced study power" appears to conflate traditional confirmatory hypothesis testing with Multiverse Analysis. The manuscript employs a pre-registered multiverse framework (OSF), which fundamentally mitigates the risks of p-hacking or post-hoc data dredging. By testing 10 pre-specified compliance (CACE) models across distinct theoretical dimensions (such as fidelity, dosage, and reach), the authors are not "cherry-picking" a p-value; rather, they are mapping the robustness and stability of the intervention effect against the inherent "researcher degrees of freedom" that often plague school-based trials.

From a causal inference perspective, defining a "complier" in a real-world educational setting is a notorious methodological challenge. The authors' decision to apply multiple thresholds (50th and 75th percentiles) aligns with cutting-edge recommendations from scholars like Peugh et al. (2017) and institutions such as the Education Endowment Foundation. This transparency is essential for validating the exclusion restriction assumption of the CACE model, ensuring that the findings regarding the "Passport" intervention are not merely artifacts of a single, arbitrary statistical cutoff.

Ultimately, this work represents an exemplary application of rigorous, open science that fits perfectly within the scope of PLOS ONE. It provides a valuable contribution to implementation science by demonstrating how varying compliance metrics can shift, or confirm, an effect.

Response: We extend our sincerest thanks to Reviewer 2 for their encouraging feedback on our manuscript.

Reviewer #3

1. It is noted that the statistical analysis of the tables attached in Supplementary File 2 was not adequately addressed. For example the significance of the R2 value, its significance, and the justification for its decrease in some cases. Likewise, the analysis of what was stated in Supplementary File 3 suffers from a lack of depth in expansion and statistical significance.

Response: File S1 - We have changed R2 to lamba in File S2. We do not discuss the variation in this value as it does not impact our models or results, nor is it relevant to our research question. As this manuscript is already of considerable length, with detailed discussion of model variations, we feel that further discussion of this offers little value to the reader.

File S2 - This file has 12 tabs detailing the output of the multiverse analysis. Statistical significance is denoted by an asterisk, which is explained under each table. The model results, and variation in results, are discussed in depth throughout the results and discussion sections of the manuscript.

For example, we examine, identify and discuss the presence and absence of statistically significant compliance predictors across models (page 27 line 609-622) and we discuss the implications of this finding for future research (page 33 lines 774-782).

To improve clarity within the tables, we have changed the title of the teacher-level variables – for example ‘SEL’ has been changed to ‘Attitudes towards SEL’.

2. It was suggested that a 50% threshold would suffice for the compliance standard.

Response: This misunderstanding stems from a grammatical error, as our pre-registration states that a 50% and 75% threshold would be implemented. For clarity, we have changed the order of sentences in the Introduction (page 10 lines 230-234) to read: Researchers are encouraged to use multiple sensitivity analyses to observe any change across compliance thresholds [46]. Using the 50th and 75th percentile to define compliance is established practice in educational research [41, 49, 50, 57, 58].

3. I believe that analyzing sensitivity is appropriate for the study.

Response: We appreciate that the reviewer agrees with our analysis – thank you.

---

## [Decision Letter · Decision Letter 2]

8 May 2026

CACE closed: A multiverse examination of the influence of implementation variability on student outcomes in a randomised controlled trial of a universal, school-based social-emotional learning intervention

PONE-D-25-61187R2

Dear Dr. O'Brien,

We’re pleased to inform you that your manuscript has been judged scientifically suitable for publication and will be formally accepted for publication once it meets all outstanding technical requirements.

Kind regards,

Nate Breznau

Academic Editor

PLOS One

Additional Editor Comments (optional):

Reviewers' comments:

Reviewer's Responses to Questions

**Comments to the Author**

1. If the authors have adequately addressed your comments raised in a previous round of review and you feel that this manuscript is now acceptable for publication, you may indicate that here to bypass the “Comments to the Author” section, enter your conflict of interest statement in the “Confidential to Editor” section, and submit your "Accept" recommendation.

Reviewer #1: All comments have been addressed

2. Is the manuscript technically sound, and do the data support the conclusions?

Reviewer #1: Yes

3. Has the statistical analysis been performed appropriately and rigorously? 

Reviewer #1: Yes

4. Have the authors made all data underlying the findings in their manuscript fully available?

Reviewer #1: Yes

5. Is the manuscript presented in an intelligible fashion and written in standard English?

Reviewer #1: Yes

6. Review Comments to the Author

Reviewer #1: I thank the authors for the thoughtful responses and thorough revisions to the feedback. It is my opinion that the manuscript is ready for publication in PLOS One.

7. PLOS authors have the option to publish the peer review history of their article (what does this mean?). If published, this will include your full peer review and any attached files.

Reviewer #1: No

---

## [Editor Report · Acceptance letter]

PONE-D-25-61187R2

PLOS One

Dear Dr. O'Brien,

I'm pleased to inform you that your manuscript has been deemed suitable for publication in PLOS One. Congratulations! Your manuscript is now being handed over to our production team.

Kind regards,

on behalf of

Dr. Nate Breznau

Academic Editor

PLOS One